# The Use of Wearable Sensors and Machine Learning Methods to Estimate Biomechanical Characteristics During Standing Posture or Locomotion: A Systematic Review

**DOI:** 10.3390/s24227280

**Published:** 2024-11-14

**Authors:** Isabelle J. Museck, Daniel L. Brinton, Jesse C. Dean

**Affiliations:** 1Department of Health Sciences and Research, Medical University of South Carolina, Charleston, SC 29425, USA; 2Department of Healthcare Leadership and Management, Medical University of South Carolina, Charleston, SC 29425, USA; brintond@musc.edu; 3Department of Rehabilitation Sciences, Medical University of South Carolina, Charleston, SC 29425, USA; 4Ralph H. Johnson VA Medical Center, Charleston, SC 29401, USA

**Keywords:** balance, biomechanical characteristics, center of pressure, machine learning, postural mechanics, wearable sensors

## Abstract

Balance deficits are present in a variety of clinical populations and can negatively impact quality of life. The integration of wearable sensors and machine learning technology (ML) provides unique opportunities to quantify biomechanical characteristics related to balance outside of a laboratory setting. This article provides a general overview of recent developments in using wearable sensors and ML to estimate or predict biomechanical characteristics such as center of pressure (CoP) and center of mass (CoM) motion. This systematic review was conducted according to PRISMA guidelines. Databases including Scopus, PubMed, CINHAL, Trip PRO, Cochrane, and Otseeker databases were searched for publications on the use of wearable sensors combined with ML to predict biomechanical characteristics. Fourteen publications met the inclusion criteria and were included in this review. From each publication, information on study characteristics, testing conditions, ML models applied, estimated biomechanical characteristics, and sensor positions were extracted. Additionally, the study type, level of evidence, and Downs and Black scale score were reported to evaluate methodological quality and bias. Most studies tested subjects during walking and utilized some type of neural network (NN) ML model to estimate biomechanical characteristics. Many of the studies focused on minimizing the necessary number of sensors and placed them on areas near or below the waist. Nearly all studies reporting RMSE and correlation coefficients had values <15% and >0.85, respectively, indicating strong ML model estimation accuracy. Overall, this review can help guide the future development of ML algorithms and wearable sensor technologies to estimate postural mechanics.

## 1. Introduction

Balance deficits are common in many clinical populations, specifically among individuals with lower extremity musculoskeletal injuries [1], stroke [2], degenerative diseases [3], and advanced age [4]. An estimated 13% of adults between ages 65 and 69 self-report imbalance and this proportion increases to 46% in those aged 85 and older [5]. Additionally, balance issues are often precursors of falls [6], which are the leading cause of injury-related mortality among older adults in the United States [7]. Deteriorating control of balance can greatly impact functional mobility [8], quality of life, and independence [9].

Biomechanical measures of gait and standing posture can assess a patient’s balance function and provide insight into potential rehabilitation goals or treatments [10]. The current gold standard systems used to evaluate biomechanical parameters of gait and postural balance, such as force platforms and motion capture systems, are complex, expensive, and often require highly trained operators, preventing translation outside the laboratory and clinic [11,12]. These systems often quantify characteristics of body motion based on the trajectory of the center of pressure (CoP) or center of mass (CoM) [12]. These biomechanical characteristics can provide insight into the underlying balance control during either standing or walking and can be used to assess the effects of rehabilitation interventions [13,14]. 

Recently, substantial work has focused on the development of portable technology that allows the measurement of biomechanical characteristics outside of a laboratory setting. Wearable technologies offer a low-cost, portable way to track biomechanical characteristics in real-time [12], with typical wearable sensors such as inertial measurement units (IMUs) including accelerometer, gyroscope, and magnetometer components. Several systematic reviews have addressed the use of wearable devices to track biomechanical characteristics and estimate fall risk for older adults. A systematic review from 2017 investigated the use of mobile phone technology to assess balance and fall risk [15], while another from 2018 addressed different novel sensing technologies used for fall risk assessment in older adults [16]. More recently, a review from 2019 focused on providing a detailed overview of the common applications of wearable sensors to assess postural balance [12]. The use of wearable sensors to assess balance, however, is an area of research that is continually evolving. While existing reviews focus on using data collected directly from wearable sensors to assess balance or fall risk, there is a noticeable gap in research evaluating the use of wearables to estimate or predict balance characteristics.

The multi-dimensional, multi-correlated, and patient-specific nature of the data collected from postural and gait analysis can challenge analysis and comprehension. One possible approach to overcoming this challenge may be to use machine learning (ML) and artificial intelligence (AI) methods to process data to predict or estimate biomechanical characteristics. Prediction involves forecasting future or unseen data values from previous training data while estimations approximate current data or parameters based on historical data. In the healthcare sector, ML has rapidly gained attention due to its demonstrated potential to solve problems and perform certain medical tasks such as diagnosis and detection more efficiently and effectively than humans [17]. However, the use of ML models to assess human gait and postural balance is a relatively new area of research [18]. While such ML applications were briefly discussed in the 2019 review of wearable sensors [12], there is a lack of an updated and comprehensive overview addressing the integration of wearable sensors and machine learning technology to estimate or predict continuous biomechanical characteristics such as CoP and CoM motion. 

The purpose of this research is to provide an overview of recent work that has incorporated wearable sensors with machine learning technology to estimate or predict biomechanical characteristics (CoP and CoM) while standing or walking. We thus extend upon the previous systematic reviews published in the last few years that focused on more traditional approaches to IMU-based balance assessment. This work provides a detailed review of emerging research and technologies that will likely be a component of future personalized, adaptive, and portable rehabilitation therapies for those with balance, posture, and gait abnormalities. Through the selection of high-quality papers that combine machine learning technology and wearable sensors to quantitatively estimate biomechanical characteristics, this study aims to (1) investigate optimal machine learning models for estimating CoP and CoM from wearable sensor data inputs, (2) identify common locations for sensor placement and testing conditions, (3) identify common ML model performance metrics across studies, and, finally, (4) suggest future directions of design and implementation of wearable sensors and ML models to accurately estimate and predict postural mechanics. This review is organized into five sections: Section 1 provides the background and rationale for the study; Section 2 outlines the methodology for searching, selecting articles for inclusion, extracting data, and assessing quality; Section 3 presents the results and characteristics of the included articles; Section 4 discusses these findings and the study limitations; and Section 5 concludes with the implications and future directions of this research.

## 2. Materials and Methods

### 2.1. Search Strategy

A literature search was conducted using the following six databases: Scopus, PubMed, CINAHL, Trip PRO, Cochrane, and OTseeker. Articles including a variation of the following terms were included: “artificial intelligence algorithms”, “wearable sensors”, and “biomechanical characteristics”. Title and abstract keywords and their synonyms were converted from the different database search syntaxes accordingly. The interest in ML for healthcare has increased rapidly over the last 10 years. Following 2010, the number of clinical practice and clinical research-related publications with machine learning increased exponentially [19]. To evaluate recent trends, only articles published from January 2010 to February 2024 were included in this review. This search includes English-language, peer-reviewed journal articles only. The computerized database search was performed using the following query in Table 1.

### 2.2. Study Selection

The method of selecting relevant articles consisted of the following process given by the Preferred Reporting Items for Systematic Reviews and Meta-Analyses (PRISMA) guidelines [20]. Details regarding the necessary reporting items can be found in the PRISMA 2020 checklist in the Appendix A. Each database was last consulted on 2 February 2024. Using the EndNote reference manager software version 21, duplicates were removed, and the remaining articles were hand-screened by their titles and abstracts. To reduce the risk of bias, a random selection of 50 articles from the original search (with duplicates removed) was sent to an outside reviewer for additional title and abstract screening. Following title and abstract screening, articles were read fully and filtered out by the primary reviewer according to the following inclusion and exclusion criteria. 


*Inclusion Criteria:*
Only peer-reviewed, web-available journal articles.Articles published in the period from January 2010 to February 2024.Studies that use wearable sensor data as inputs for model prediction.Articles using machine learning to estimate or predict CoP, CoM, or a variant of these two balance-related biomechanical characteristics.



*Exclusion Criteria:*
Reviews, conference papers, magazines, or book chapter papers.Studies in languages other than English.Publications that studied robotic, exoskeleton-based systems, or non-human subjects.Studies that used non-wearable sensors-based systems, such as cameras and radar to make predictions.Studies that used ML and wearable sensors solely to detect instability, falls, freezing of gait, or assess fall risk.Studies that used ML and wearable sensors to classify different postural behaviors or postural statuses.Studies that used ML and wearable sensors to estimate PT labels or clinical rating scales of balance performance.Studies that used ML and wearable sensors solely to predict joint angles, torques, and ground reaction forces (GRFs).


### 2.3. Data Extraction

After confirming with a collaborator, the data items needed for meaningful synthesis were collected from each report. Study objectives, participant health status, age, sample size, and level of evidence were all extracted from the selected articles. Additionally, to facilitate analysis and investigate the current state of using machine learning models and wearable sensors to estimate biomechanical characteristics, the following information was extracted from the selected articles: (1) Machine learning model-related information: including the type of model used, features extracted, and training method. (2) Sensor-related information: including the sensor type, number, and locations during signal acquisition. (3) Testing condition information: data were acquired under various testing conditions (walking, static standing, and running). (4) Duration of collected biomechanical data, as in the duration of the trials for each testing condition. (5) Biomechanical characteristic of interest: different studies focused on the estimation and prediction of several different biochemical parameters including the center of pressure (CoP), the center of mass (CoM), the center of mass height (CoMH), the center of pressure–center of mass inclination angle (CoP-CoM IA), and the strike index. The strike index is defined as the location of the center of pressure along the long axis of the foot as a percentage of the total foot length at the initial foot strike [21]. Different outcome-related information such as the error ratio (ER), root-mean-square errors (RMSEs), normalized root-mean-square errors (NRMSEs), Jaccard index [22], and R-squared values were used to report the performance of the machine learning model predictions and estimations. The primary outcomes sought from each study were these performance metrics demonstrating the ML model’s ability to predict characteristics like CoP or CoM displacement, velocity, location, height, or angle. The extracted data were tabulated to help compare study properties and results. Missing summary or unclear data were analyzed by the reviewer and reasonable inferences were made based on accompanying information. 

### 2.4. Quality Assessment

Each article was screened and scored using the Downs and Black Quality checklist to assess overall methodological quality, internal and external validity, and power of the study [23]. The Downs and Black checklist consists of 27 items distributed between 5 sub-scales: reporting, external validity, bias, confounding, and power. The articles were scored Yes = 1, No = 0, and Unclear/NA = 0 for each item on the checklist. Investigating the differences in sample and study characteristics such as subject age, health condition, type of ML model applied, test condition, and estimated biomechanical characteristic allowed for exploration of possible causes of heterogeneity among the study results. Studies with similar samples, characteristics, and outcome measures were grouped for comparison and analysis. The level of evidence of each study was also investigated to assess confidence and certainty in the results. Each study design was categorized based on a recently developed pyramid for classifying AI scientific evidence presented by Bellini and colleagues shown in Figure 1 [24]. 

## 3. Results

A detailed flow diagram of the search results and screening strategy is presented in Figure 2. The initial search resulted in 192 articles for possible inclusion in the present review. After the removal of duplicated studies, 131 articles remained for title and abstract screening. The initial screening of titles and abstracts removed 96 studies, and the remaining 35 articles were reviewed in their full-text versions to assess their inclusion in the review. The full-text screening removed 21 studies due to the previously stated exclusion criteria, which involved (i) studies that used IMUs and ML to classify postural status (such as sitting, standing, and walking) [25,26], (ii) studies that focused on classifying postural behaviors [27,28,29,30,31] or predicting joint kinetics [32], (iii) studies that assessed fall risk or detected falls [33,34], (iv) studies that estimated clinical scales or PT ratings [35,36,37,38] (v) studies that used non-human subjects [39,40,41], (vi) and studies that were not peer-reviewed [42,43,44,45]. The rigorous review process resulted in 14 quality articles that were included in this systematic review.

Table 2 summarizes the data extracted from the 14 articles, including the author and year of publication, characteristics of the study population, testing conditions, machine learning models applied, estimated biomechanical characteristics, and sensor-related information. The study type, level of evidence, and the Downs and Black scale score out of 27 points are also reported in Table 2.

### 3.1. Study Design and Methodological Quality 

The distribution of methodological quality and bias assessment scores based on the Downs and Black checklist are presented in Table 2. The reviewed articles had an average Downs and Black score of 15 with scores ranging between 11 and 21, and over half of the scores ranging between 13 and 15 points. All the studies, with the exclusion of one [48], were considered internal validation studies according to the AI scientific evidence pyramid [24] and were classified as level III evidence using Sackett’s criteria [58]. 

### 3.2. Outcome Measurements

There are several different methods by which the studies in this review reported results and outcomes of ML model predictions. The results from each study quantifying the ML model’s accuracy in predicting the biomechanical characteristic of interest (CoP, CoM, CoP-CoM IA, etc.) are represented in Table 3**.** Current ML-based IMU literature widely uses the root mean square error (RMSE) to evaluate the prediction accuracy of the estimated gait variables [55]. All studies, except three [47,48,50], reported prediction outcomes in terms of RMSE and/or normalized root-mean-square error (NRMSE). RMSE is reported in units of the response variable, or biomechanical characteristic in this case, and is calculated by taking the square root of the average of the squared differences between the true or actual values and the predicted values. The NRMSE is simply the RMSE divided by the range of values and is reported as a percentage. Lower RMSE and NRMSE values indicate higher model prediction accuracy. Both measurements are useful for comparing different models with one another. Many of the studies (64%) used a measure related to correlation coefficient such as R-squared or Spearman’s Rho [47] to evaluate a model’s ability to estimate gait or standing biomechanical characteristics. Correlation coefficients are useful in determining whether a model is objectively a good fit for the data. Coefficients close to unity (1.0) indicate strong relationships and lower model errors. One study [50] reported outcomes as an error ratio (ER) percentage of the estimated CoP and true CoP, while another study [54] reported model prediction results in terms of the Jaccard Index, which is a measure of similarity between two datasets determined by calculating the percentage of the size of the intersection between ground truth and predicted CoP (the number in both sets) divided by the size of union between the two (the number in either set). Although the performance of the models in each of the studies can be evaluated individually, it is difficult to compare the performance of these models across studies due to the differences in various parameters such as inputs, outputs, validation, and model training data. 

### 3.3. Machine Learning Models

The 14 reviewed studies applied a variety of ML models to estimate biomechanical characteristics from collected wearable sensor data. Many of the studies applied and tested more than one ML model. A large majority of the machine learning applications (about 77%) were neural network (NN) models. Neural networks are a type of ML model that mimics biological neuroscience and uses layers of interconnected neurons to learn from examples and make decisions, recognize patterns, or solve problems [59]. More specifically, of the 22 applications of machine learning, 7 studies applied a type of long short-term memory (LSTM) modeling. LSTM models are a type of neural network that uses memory cells to learn long-term dependencies in sequential data. This structure makes such models suitable for time series predictions and estimating changing parameters like CoM and CoP [56]. Each model’s frequency of use in the reviewed studies is summarized in Figure 3. 

### 3.4. Wearable Sensor Properties

Two types of sensors were primarily used in these studies: inertial measurement units (IMUs) and triaxial accelerometers. Almost 80% of the studies used IMU sensors, which are typically composed of accelerometer, magnetometer, and gyroscope components that capture motion signals. There was one study [46] that used IMUs in combination with force-sensing resistors (FSRs) as input for ML predictions. In terms of sensor location, the lower body including the feet, shanks, thighs, and hip/lower back region were the most common locations to which sensors were attached. There is a variety in the number of sensors used to make predictions among the studies in this review; however, many studies had the stated goal of minimizing the number of sensors. Figure 4 indicates the locations of sensor placement and Figure 5 summarizes the characteristics of the wearables used in conjunction with the ML models to predict biomechanical characteristics in the reviewed articles.

### 3.5. Testing Activity

Walking (60%) was the primary testing activity used to explore the use of ML models and wearable sensors to predict biomechanical characteristics. Standing balance (33%) was another common condition tested. This was often not just simple, static, bipedal standing but rather standing on one foot or in tandem [47], standing while leaning in the AP/ML directions [50], or standing while receiving external perturbations [48,52]. Only one article [21] focused on the estimation of biomechanical characteristics during running. The different test activities may cause differences within the results from these studies and can make it difficult to make direct comparisons as prediction accuracy may be affected by varying activities like standing, walking, and running.

### 3.6. Predicted Biomechanical Characteristic of Posture

Half of the studies (50%) in this review focused on using ML models and wearable sensors to predict or estimate a subject’s center of pressure (CoP). Five of the studies focused on the prediction of certain center of mass measurements (CoM) such as CoM motion, CoM height, or CoM acceleration, and two of the studies involved the incorporation of both CoP and CoM characteristics to predict the CoP-CoM inclination angle (CoP-CoM IA). The prediction of different biomechanical characteristics is an additional cause for variations between the results of these studies. 

## 4. Discussion

This systemic review examined literature that estimated biomechanical characteristics such as CoP and CoM using machine learning, with a focus on the integration of sensor technology and ML models. There were a variety of patient populations, testing protocols, and characteristics examined among the studies included in this review. Although multiple different types of ML models and sensor configurations were used to make predictions across studies, nearly all the authors reported favorable ML model prediction and estimation outcomes. 

### 4.1. Participant Characteristics

Almost all the papers included in this review involved healthy individuals in their mid-twenties, while few included participants with a neurodegenerative condition such as stroke or a spinal cord injury. Many previous studies involving the initial investigation of the integration of ML models and wearable sensors similarly used healthy or able-bodied subjects in their studies [60]. It is common to begin ML model exploration with healthy control subjects and then move into more complex systems such as patients with different neurological conditions. In a recent study investigating the use of ML models for gait activity classification, Slemenšek and colleagues first focused on identifying the best-performing ML algorithm for activity classification of a single 25-year-old male subject and then expanded to a larger dataset of five subjects, with a final follow-up proof of concept on patients with neurological pathologies [61]. The goal of much of this research is to ultimately develop models to help those with gait or balance impairments; therefore, the articles in this review that tested their ML models on both healthy controls and pathological subjects [46,49,57] are of additional value.

### 4.2. Testing Parameters and Sensor Properties

The task performed by participants can have a considerable effect on the accuracy of ML model predictions, as can the number and location of wearable sensors. Sensor number and placement varied from study to study; however, the most frequent sensor placement locations across all the studies were the feet, shanks, thighs, and hip/lower back region. This is consistent with the most common sensor placement locations from a recent systematic review of wearable sensor-based technologies for fall risk assessment [62]. Vincent et al. demonstrated that the location (upper trunk, waist, lower thigh) of an accelerometer influenced the accuracy of CoP estimation during standing, regardless of the different ML models applied [50]. Of the three separate locations they tested in the study, it was determined that the waist was the optimal position for sensor placement to estimate CoP while standing. One of the studies in this review tested model performance with multiple different sensor locations and found that when using one IMU for estimation of CoP during walking, accuracy levels ranked from best to worst were back, right thigh, left thigh, left shank, right shank, right foot, and left foot [49]. Similarly, Howcroft et al. determined that sensor data collected from areas near the center of mass (i.e., lumbar and waist) were most sensitive and performed the best for identifying fall risk compared with other locations [63]. A separate study [61] determined sensor placement below the knee joint, anterior of the shin bone was superior to more proximal placement in terms of differentiating different gait activities rather than estimating a continuous parameter like CoP or CoM. The more noticeable differences in lower leg movements during different gait activities likely contributed to making these classifications more distinguishable. Overall, it seems that sensors placed on the lower half of the body perform the best in predicting CoM and CoP, and placement near the sacrum or waist is particularly sensitive to postural changes, making for more accurate estimations of these balance characteristics. 

Across the studies, participants performed a variety of different tasks during data collection such as standing still, standing on one foot, walking, and even running. The chosen placement location of the sensors is likely largely dependent on the task performed during data collection. For example, it is reasonable to assume that assessments of standing balance would benefit little from IMU sensor data collected from the feet, as there is little to no foot movement during static standing. On the other hand, sensor data collected from the feet during walking could be very valuable input for ML model predictions. 

Similar to the review from Jiao and colleagues on automatic post-stroke classification systems [64], the studies in this analysis used different numbers of sensors to train ML model predictions. Although, in theory, more sensors lead to more data and could lead to more accurate ML predictions, it was more common for the articles in this review to investigate the use of 3 or fewer wearable sensors with promising ML model prediction accuracy. This is in line with current literature that has demonstrated over 90% accuracy in gait classification and gait phase estimation using a single IMU sensor and ML models [65]. One of the studies included in this review [49] tested their ML models with differing quantities of IMU sensors and found that the number of IMUs can be reduced to five without deterioration in the model performance. Although fewer sensors are ideal as less equipment is more easily translated to real-world settings, researchers must be careful to select the proper number of sensors to ensure their desired prediction accuracy. 

### 4.3. Limitations

Small sample sizes contribute to a risk of inaccurate results for many of the reviewed studies, which often recruited fewer than 25 participants for ML model development and even fewer for those that included stroke patients. One study reported data from a single participant [48]. Although some studies collected multiple trials from each participant to expand the data set, this way of model training may have caused overestimations of classification performance due to decreased variation in the training and test data [64]. Additionally, the low number of both healthy and pathological participant observations may have affected the ML model development and its ability to generalize to new data. 

Differences in analysis methodology also complicate comparisons between the identified studies. The most common statistical measures for evaluating the performance of regression are Mean Absolute Error (MAE), Mean Squared Error (MSE), root-mean-square error (RMSE), R^2^, and adjusted R^2^ [66]. Some studies only reported one of these outcomes [47,48] and others included other metrics of reporting accuracy such as the Jaccard Index and Error Ratio [50,54], preventing direct comparison of model performance among studies. Standardization among reporting ML model performance would allow models from different studies to be compared more accurately. However, the varying goals of these models may prevent agreement on the most appropriate metric. Other factors such as differences in training data, validation processes, sample sizes, and hyperparameters can also affect the outcome metrics across the models in each study. Thus, future models could include a standard for training data set size and percentage of validation and test data to allow for consistency and valid comparison amongst ML model results. Nearly all of the studies that reported correlation coefficients and NRMSE, however, had values > 0.85 and <15%, respectively, indicating that most of the models across all the studies in this review demonstrated strong performance in estimating CoP and CoM. Liu and colleagues proposed that a machine learning model is highly accurate if the correlation coefficient is greater than 0.9 and the NRMSE is 15% or less [67]. The required model performance will depend on its application within the field. For example, a model used to detect cancer from MRI images will likely require higher prediction accuracies for clinical adoption than a model used to predict time in surgery for orthopedic patients. Although these results are promising and valuable for guiding the future development of ML models to predict CoP or CoM, it is important to consider internal and external validation methods, the size and diversity of the training data set, and the generalizability of these results to truly assess their efficacy and performance in a clinical setting and as a rehabilitation intervention. Further research is necessary for all models tested in these studies to properly demonstrate efficacy with larger sample sizes and more diverse populations. 

This systematic review provides an overview of the recent literature on the development of ML technology to predict CoP and CoM using wearable sensors. Future research could involve a more detailed data extraction process from the ML models included in each study to allow for an even deeper understanding of the model performance and a stronger comparison of the results. Extracting details regarding the training data, test data, validation process, and model hyperparameters would allow for a more detailed and comprehensive review of ML model development and performance among these studies. 

## 5. Conclusions

The results from this review indicate that ML models can accurately estimate CoP and CoM data from wearable sensor data. Despite the challenge of identifying a single optimal model due to the diverse characteristics of the studies reviewed, neural networks were the most commonly used model and have demonstrated highly accurate predictions. Most studies focused on estimating biomechanical characteristics during walking, with sensors typically placed on the feet, hips, and lower limbs. Model performance metrics were often reported as correlation coefficients or RMSE values. 

Although a variety of models, testing parameters, sensors, and validation methods were used among the studies in this review, this information can be used to guide new studies in the development of optimal methods for designing and testing ML models that make estimations from wearable sensor data. Future research may focus on a more detailed analysis of machine learning model parameters and the development of standardized methods for stronger comparisons of model performance across studies. Moving forward, researchers can aim to enhance the design of these models by creating systems capable of making accurate predictions across various daily activities. Additionally, reducing the number of sensors required for these predictions will make the system more portable and user-friendly. Overall, the results of this review will help direct the development of novel methods, devices, and models for balance rehabilitation. 

## Figures and Tables

**Figure 1 sensors-24-07280-f001:**
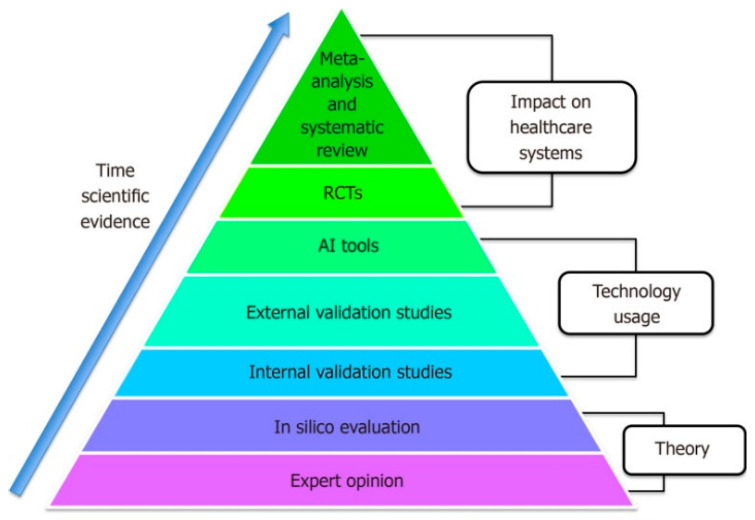
Pyramid for artificial intelligence scientific evidence [24]. *Adapted from* “*The artificial intelligence evidence-based medicine pyramid,” by Bellini, V., et al., 2023, World J Crit Care Med, 2023. 12(2): p. 89–91. Copyright 2023. Reprinted with permission*.

**Figure 2 sensors-24-07280-f002:**
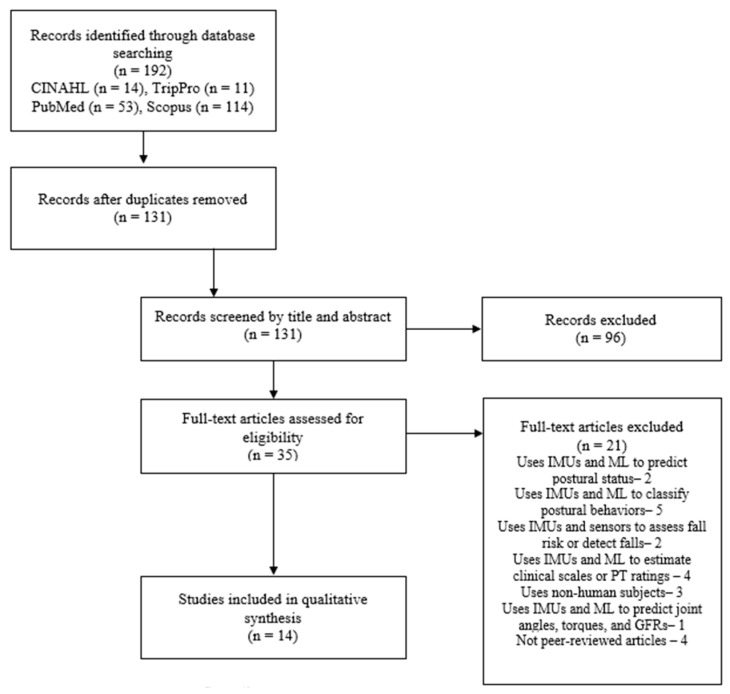
PRISMA flow diagram.

**Figure 3 sensors-24-07280-f003:**
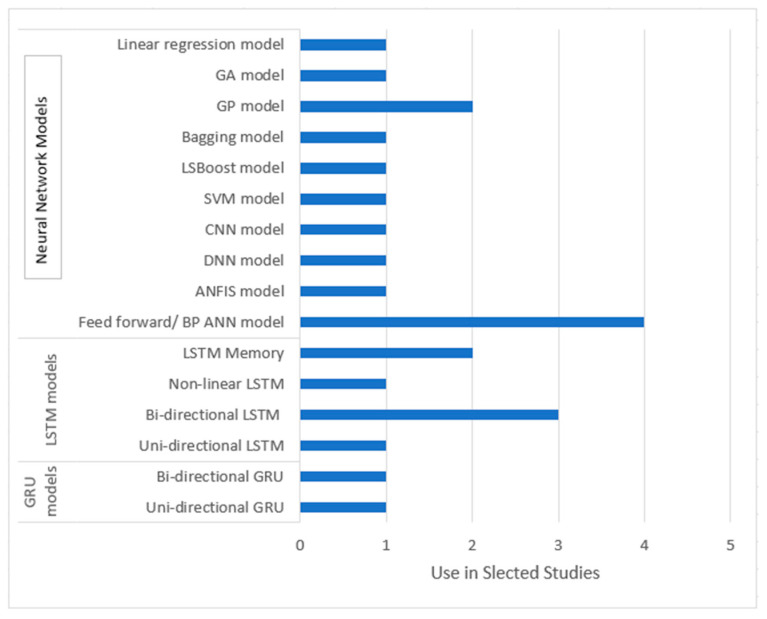
Summary of the different ML models applied in the reviewed studies.

**Figure 4 sensors-24-07280-f004:**
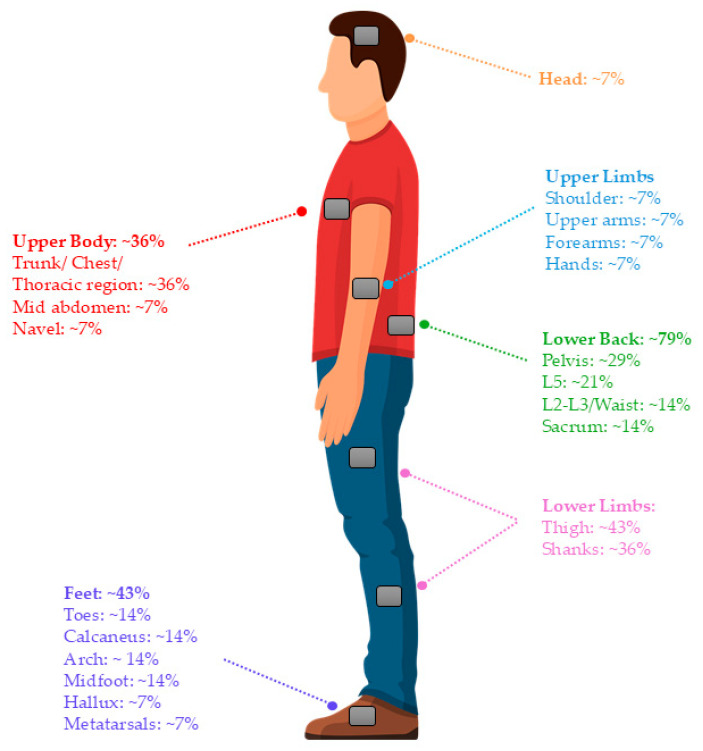
Sensor placement locations and the percentage of studies included in this review that consider each position (percentages are based on counts of studies that included that specific sensor location and may not add up to 100%).

**Figure 5 sensors-24-07280-f005:**
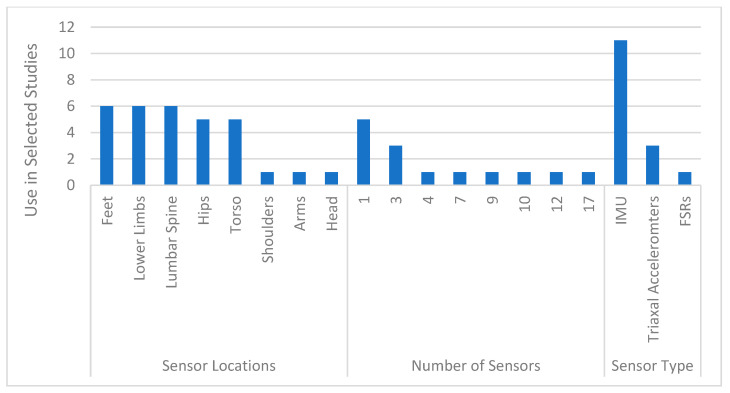
Wearable sensor characteristics.

**Table 1 sensors-24-07280-t001:** Systematic review search query.

Parameter	Search Query
Artificial intelligence algorithm	(“artificial intelligence” OR AI OR “machine learning” OR ML OR “long short-term memory” OR LSTM OR “artificial neural network” OR ANN OR “neural network” OR “support vector machine” OR SVM) AND
Wearable sensor	(“wearable sensor” OR IMU OR “inertial measurement unit” OR accelerometer OR gyroscope OR magnetometer) AND
Biomechanical characteristics	(“center of pressure” OR COP OR “center of mass” OR sway)

**Table 2 sensors-24-07280-t002:** Summary of study characteristics. UND: trial duration for collected biomechanical data was not reported. CoP: center of pressure; CoM: center of mass; LSTM: long short-term memory; IMU: inertial measurement unit.

First Author (Year)	Study Design	Level of Evidence	Downs & Black Scale Score	Participant Characteristics	Testing Condition	Data Collection Trial Duration	Estimated Mechanical Parameter	ML Method Used	Wearable Sensors Used and Locations
**Yu, C. H. (2023) [46]**	Internal Validation study	III	21	26 healthy individuals, 13 subjects with 25.46 +/− 2.37 years of age and 13 subjects with 72.75 +/− 6.68 years of ages	Walking	UND	CoM-CoP inclination angle (IA)	4 Types of Recurrent Neural Network (RNN) models: unidirectional-LSTM, bi-LSTM, uni-GRU, and bidirectional -GRU	1 IMU; located on sacrum
**Duong, Ton T. H. (2023) [47]**	Internal Validation study	III	15	38 (13 healthy individuals and 25 with genetically determined neuromuscular conditions), 10.1–29.8 years	Walking	6 min	CoP	Bidirectional-LSTM model and Gaussian Process Regression (GP)	8 FSRs on the foot and 1 IMU located under the midfoot
**Labrozzi, Gabrielle (2023) [13]**	Internal Validation study	III	15	5 healthy individuals, 24–37 years	Walking and standing balance	UND	CoM	Bi-directional LSTM model	12 IMUs; three located on the thoracic region, three on the mid-abdomen, one positioned anterolaterally on each thigh and shank, and one per foot
**Wantanajittikul, Kittichai (2022) [48]**	Internal Validation study	III	18	53 healthy individuals, an average of 46 years	Standing balance	30 s	CoP	Least-square boosting (LSBoost), Bootstrap aggregation (Bagging), Support vector machine (SVM), Artificial neural network (ANN), and Gaussian process (GP)	1 IMU; located on the lumbar region of the trunk
**Tan, T. (2022) [21]**	Internal Validation study	III	15	16 healthy individuals, 23.0 +/− 2.2 years	Running	100 steps	Strike Index (CoP)	Convolutional Neural Network (CNN)	1 IMU sensor; located on dorsum surface of shoe
**He, B. (2022) [49]**	Internal Validation study	III	14	11 subjects (10 healthy individuals, 1 subject 8 months post-stroke), 22.7 +/− 3.6 years and 64 years old	Walking	60 s	CoM Height (CoMH)	Back propagation (BP) neural network and GRU network (a variant of recurrent neural network (RNN))	3 IMUs; located on the calf, thigh, and pelvis
**Chebel, E. (2021) [50]**	Internal Validation study	III	17	22 healthy individuals, 22–26 years	Walking and squat tasks	UND	CoM	Deep Neural Network (DNN)	5 IMUs; located on lumbosacral joint, right/left shoulders, and right/left hips
**Hnat, S. K. (2021) [51]**	Internal Validation study	III	14	5 healthy individuals, Median age of 23 years	Standing balance with internal and external perturbations	UND	CoM	Artificial Neural Network (ANN)	10 triaxial accelerometers; located on the torso, sternum, navel, pelvis, anterior thighs, and anterior shanks
**Lee, M. (2020) [52]**	Internal Validation study	III	15	20 healthy individuals, 24.7 +/− 3.2 years	Walking	2 min	CoP	Artificial Neural Network (ANN)	1 IMU; located on the Sacrum
**Wu, C. C. (2020) [53]**	Internal Validation study	III	14	5 healthy individuals, 25 +/− 1.87 years	Walking	74 steps	CoP	LSTM model	4 IMUs; located on left toe, lateral and heel parts of shoe, and waist
**Podobnik, J. (2020) [54]**	Internal Validation study	III	12	6 subjects (4 healthy individuals, 2 stroke patients), 32-55 years	Walking	30 sec standing; 3 and 7 min walking trials	CoP	Non-linear Long-Short-Term Memory (LSTM) model	7 IMUs; located on the pelvis, each thigh, shank, and foot
**Choi, A. (2019) [55]**	Internal Validation study	III	14	24 healthy individuals, 26.2 +/− 1.5 years	Walking	UND	CoM-CoP IA	Feed forward ANN model and LSTM memory model	1 IMU; located on lumbar spine
**Chen, Vincent (2018) [56]**	Internal Validation study	III	15	10 healthy individuals, 24–31 years	Standing balance	40 s	CoP	Neural network (NN), genetic algorithm (GA), and adaptive network-based fuzzy inference system (ANFIS)	3 accelerometers; located on the upper trunk, waist, and lower thigh
**Nataraj, Raviraj (2012) [57]**	Case study	IV	11	1 individual with a thoracic-level spinal cord injury, Age Unknown	Standing balance	UND	CoM acceleration	Linear regression model	2 (3-D) accelerometers; located on the pelvis and torso

**Table 3 sensors-24-07280-t003:** Summary of study results. AP: anterior–posterior, ML: medio–lateral, [H]: healthy controls, [Patho]: pathological subjects, GP: Gaussian process model; Bi-LSTM: bi-directional long short-term memory model; Bagging: bootstrap aggregation; LSBoost: least-square boosting; SVM: support vector machine; ANN: artificial neural network; GA: genetic algorism, ANFIS: adaptive network-based fuzzy inference system; DNN: deep neural network, Uni-LSTM: uni-directional long-short-term memory model; Uni-GRU: uni-directional gated recurrent unit; Bi--GRU: bi-directional gated recurrent unit; CNN: convolutional neural network; GRU: gated recurrent unit; BP neural network: Back propagation training neural network.

First Author (Year)	Testing Condition	Estimated Mechanical Parameter	Machine Learning Model	Root-Mean-square Error (RMSE) [cm or degrees]	Normalized RMSE (NRMSE) [%]	Spearman’s Rho Correlation Coefficient	Correlation Coefficient	Error Ratio (ER) [%]	Jaccard Index [%]
**Yu, C. H. (2023) [46]**	Walking	CoP-CoM Inclination Angle (IA)	Uni-LSTM	AP: ~0.75 deg	AP: ~4.5%	-	-	-	-
				ML: ~0.55 deg	ML: ~6%	-	-	-	-
			Bi-LSTM	AP: ~0.65 deg	AP: ~4%	-	-	-	-
				ML: ~0.50 deg	ML: ~5.7%	-	-	-	-
			Uni-GRU	AP: ~0.62 deg	AP: ~3.9%	-	-	-	-
				ML: ~0.49 deg	ML: ~5.5%	-	-	-	-
			Bi-GRU	AP: 0.61 (0.24) deg	AP: 3.82 (1.53) %	-	-	-	-
				ML: 0.46 (0.21) deg	ML: 5.33 (3.76) %	-	-	-	-
**Duong, T. (2023) [47]**	Walking	CoP	GP	AP: 1.74 (0.39) cm [H]; 1.90 (0.48) [Patho]	AP: 5.85(1.64) % [H]; 6.93 (2.32) % [Patho]	-	AP: 0.95 [H]; 0.86 [Patho]	-	-
				ML: 0.56 (0.09) cm [H]; 0.70 (0.12) [Patho]	ML: 8.94(2.34) % [H]; 9.49 (1.63) % [Patho]	-	ML: 0.39 [H]; 0.33 [Patho]	-	-
			Bi-LSTM	AP: 1.44 (0.29) cm [H]; 1.53 (0.39) cm [Patho]	AP: 4.79(1.11) % [H]; 5.59 (1.83) % [Patho]	-	AP: 0.97 [H]; 0.91 [Patho]	-	-
				ML: 0.51 (0.10) cm [H]; 0.60 (0.11) cm [Patho]	ML: 7.96(1.92) % [H]; 8.13 (1.41) % [Patho]	-	ML: 0.51 [H]; 0.47 [Patho]	-	-
**Labrozzi, Gabrielle (2023) [13]**	Walking and standing	CoM	Bi-LSTM	AP: 1.15 (0.80) cm [H]; 1.77 (0.62) [Patho]	-	-	-	-	-
				ML: 1.44 (0.65) cm [H]; 2.91 (0.62) [Patho]	-	-	-	-	-
**Wantanajittikul, K. (2022) [48]**	Standing variations (double stance, tandem, and single leg stance)	CoP	Bagging	-	-	0.8894	-	-	-
			SVM			0.8978			
			ANN			0.8921			
			GP			0.8934			
			LSBoost			UND			
**Tan, T. (2022) [21]**	Running	Strike Index (CoP)	CNN	-	6.9 (1.5) %	-	0.89 (0.09)	-	-
**He, B. (2022) [49]**	Walking	CoM Height (CoMH)	GRU	0.1408~0.1662 cm	-	-	0.906~0.922	-	-
			BP Neural Network	0.1717~0.1835 cm	-	-	0.811~0.854	-	-
**Chebel, E. (2021) [50]**	Walking and squat tasks	CoM	DNN	AP: 0.845 (1.28) cm	-	-	-	-	-
				ML: 0.603 (0.24) cm					
**Hnat, S. K. (2021) [51]**	Standing with internal and external perturbations	CoM	ANN	AP: 0.94 cm	AP: 14.6%	-	0.83	-	-
				ML: 0.60 cm	ML: 22.0%				
**Lee, M. (2020) [52]**	Walking	CoP	ANN	-	AP: 8.22 (4.96) %	-	AP: 0.86 (0.30)	-	-
					ML: 19.54 (9.59)%		ML: 0.01 (1.43)		
**Wu, C. C. (2020) [53]**	Walking	CoP	LSTM	-	AP: 5.88 (0.96) %	-	-	-	93%
					ML: 25.33 (2.35)%				63%
**Podobnik, J. (2020) [54]**	Walking	CoP	Non-linear LSTM	AP: 1.49 cm	-	-	-	-	-
				ML: 0.90 cm					
**Choi, A. (2019) [55]**	Walking	CoP-CoM Inclination Angle (IA)	LSTM	AP: 1.97 (0.81) deg	-	-	0.92	-	-
				ML: 0.82 (0.16) deg			0.96		
			ANN	AP: 3.01 (0.18) deg			0.81		
				ML: 1.27 (0.05) deg			0.87		
**Chen, Vincent (2018) [56]**	Standing while following AP/ML excursion requests	CoP	GA	-	-	-	AP: 0.95 (0.03)	AP: 8.1 (2.5) %	-
							ML: 0.96 (0.03)	ML: 7.3 (2.6) %	
			ANFIS				AP: 0.95 (0.02)	AP: 8.2 (2.3) %	
							ML: 0.96 (0.03)	ML: 7.3 (2.7) %	
			Neural Network				AP: 0.95 (0.02)	AP: 8.4 (2.3) %	
							ML: 0.96 (0.03)	ML: 7.0 (2.6) %	
**Nataraj, R. (2013) [57]**	Standing with external perturbations	CoM Acceleration	Linear Regression	-	-	-	AP: 0.972	-	-
							ML: 0.993		

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
