# Peer review of "The Use of Wearable Sensors and Machine Learning Methods to Estimate Biomechanical Characteristics During Standing Posture or Locomotion: A Systematic Review"

_sensors, 2024, doi:10.3390/s24227280_

Round 1
Reviewer 1 Report
Comments and Suggestions for Authors
1. Only three keywords; In general,four to six keywords are recommend.
2. In the article can not say ‘manuscript’, should say this research or review;
3. 2 material and methods The subheading below should be 2.1;
4. The Table form is not standardized; Many places are not standard, English writing is not standard;
5. The conclusion does not match the research objectives of the paper.
6. The manuscript only lists some problems existing in the research status without analytical methods;
7. The problems existing in the existing research should be raised, and the future development direction of the research should be raised.

Comments on the Quality of English Language
Need be improved.
Reviewer 2 Report
Comments and Suggestions for Authors
This paper reviewed articles using wearable sensors and machine learning methods to estimate biomechanical characteristics during standing posture or locomotion. There are some questions for the authors:
1. Why only 14 publications were included in this review? The review think that the corresponding research meet the inclusion may be more.
2. In Table 2, can the results of each study be supplemented? Table 3 seems the corresponding results, can they be summarized to one table together?
3. How long of data collected in each study in Table 2?
4. Can a figure be added to indicate the equipped locations of different sensors?
Reviewer 3 Report
Comments and Suggestions for Authors
The article offers a thorough overview of using wearable sensors and machine learning to estimate biomechanical characteristics in clinical populations. It follows PRISMA guidelines, covers multiple databases, and provides detailed information. The emphasis on testing during walking and efforts to minimize sensor use are noteworthy. Results show strong ML model estimation characteristics. However, including specific case studies and discussing challenges and potential solutions would enrich the review. Overall, it provides a valuable foundation for future developments in estimating postural mechanics using ML algorithms and wearable sensor technologies.
Please find the suggestions below to improve the article:
i. Please add more keywords to the abstract and sort it alphabetically.
ii. Please add a sentence to the introduction on how the paper is structured in the following sections.
iii. Please start the table from page 7. Otherwise, this page does not look good with only less information.
iv. It would be better if Table 2 is sorted by the year in descending order and then by the Downs & Black Scale Score in the same order. Please sort Table 3 by the first author year in descending order.
v. Please proofread the entire article.
Comments on the Quality of English Language
Please proofread the entire article in the revised version. Example in the abstract:
The review aims is (aims to) to provide a general overview of recent developments in using wearable sensors and ML to estimate or predict biomechanical characteristics such as center of pressure (CoP) and center of mass (CoM) motion. Almost all of the studies that reported RMSE and correlation coefficients had values <15% and 28 >0.85 (comma after this) respectively, indicating strong ML model estimation characteristics. Overall, this review can help guide future (the future) development of ML algorithms and wearable sensor technologies to estimate postural mechanics.
